# Barriers to optimal AEFI surveillance and documentation in Nigeria: Findings from a qualitative survey

**Semeeh Akinwale Omoleke** [ORCID]*, **Moyosola Bamidele, Laurent Cleenewerck de Kiev**

School of Global Health and Bioethics, Euclid University, Bangui, Central African Republic

* talk2semeeh@yahoo.co.uk

**Data Availability Statement:** The relevant data analysed during the study are presented in the published manuscript.

## Abstract

Effective spontaneous AEFI reporting is the first step to ensuring vaccine safety. Half of the global population lives in countries with weak vaccine safety monitoring systems, especially in the African, Eastern Mediterranean, and Western Pacific regions. Further, Immunisation services have been upscaled without commensurate effective AEFI surveillance systems. Hence, this study aims to comprehensively investigate the impediments to an effective AEFI surveillance system. Given the programmatic and regulatory implications, understanding these impediments would facilitate the development and implementation of policies and strategies to strengthen the AEFI surveillance system in Nigeria. A qualitative research design (using a grounded theory approach) was employed by conducting ten key informant interviews and two Focus Group Discussion sessions among the study population to identify the barriers impeding optimal AEFI surveillance and documentation in Nigeria. This study found that the AEFI surveillance system is in place in Nigeria. However, its functionality is sub-optimal, and the potential capacity is yet to be fully harnessed due to health systems and socio-ecological impediments. The identified impediments are human-resource-related issues- knowledge gaps; limited training; lack of designated officers for AEFI; excessive workload; poor supportive supervision and attitudinal issues; caregiver's factor; governance and leadership- moribund AEFI committee; lack of quality supervisory visit and oversight and weak implementation of AEFI policy guidance. Others include funding and logistics issues- no dedicated budget provision and weak referral mechanism; insecurity; socio-economic and infrastructural deficits- poverty, geographical barriers, limited ICT skills, and infrastructure; and poor feedback and weak community engagement by the health workers. Findings from this study provide empirical evidence and serve as an advocacy tool for vaccine pharmacovigilance strengthening in Nigeria. Addressing the impediments requires health system strengthening and a whole-of-the-society approach to improve vaccine safety surveillance, restore public confidence and promote vaccine demand, strengthen PHC services, and contribute to attaining UHC and SDGs.

**Funding:** The authors received no specific funding for this work.

**Competing interests:** The authors have declared that no competing interests exist.

## Introduction

Immunisation remains pivotal to public health efforts against vaccine-preventable diseases (VPDs) and emerging highly pathogenic antigens worldwide [1, 2]. There has been a significant reduction in deaths and morbidity associated with VPDs, even in low-and-middle-income countries (LMICs) that are disproportionately affected [1, 3–5]. Globally, 3.5- 5million childhood deaths are averted every year and more than 15 million future deaths have been halved with increasing access to immunisation services [6, 7]. To date, LMICs remain challenged by access to life-saving vaccines while uptake and coverage remain sub-optimal due to health system, socio-cultural factors, and geographical disadvantages and poverty [5, 8, 9].

However, vaccines are not entirely safe; they may cause mild to moderate discomfort and serious adverse events that may warrant hospitalisation or lead to death [10, 11]. Indeed, serious adverse events following immunisation (AEFI) caused by vaccines are rare (<1 in 1,000,000 cases). Despite this significant safety profile, caregivers and parents sometimes express concern and hesitancy for vaccination. The AEFI or perceived vaccine safety concern is one of the causes of vaccine hesitancy in the form of outright rejection or drop-out in many countries, including Nigeria [11–13].

Increasingly vaccines are being introduced in the expanded programme on immunisation in many LMICs, necessitating the establishment of functional surveillance systems for adverse events following immunization [14, 15]. This is ideally an integral part of national immunisation programmes. Further, the World Health Organization (WHO) Guideline on Vaccine Safety Monitoring stipulates that all countries must have the minimum capacity to detect, report, investigate and document serious and non-serious AEFI [16, 17]. The minimum reporting rate for AEFI is 10 per 100,000 surviving infants per year and many countries do not meet this requirement in Africa [18]. In 2020, two new vaccines were granted emergency use licensing (EUL), namely COVID-19 vaccines and nOPV2by the WHO [19]. These vaccines (COVID-19 vaccine and nOPV2) were allowed for public health use (introduced to millions of people) given the horrific deaths and economic losses due to the COVID-19 pandemic and explosive outbreaks of circulating vaccine-derived polio type 2 outbreak respectively. At the time of writing this manuscript (22nd August 2022), Nigeria added the Rotavirus vaccine to the national EPI schedule (making a total of 10 antigens), intending to reach 7 million children within 12 months, requiring a functional vaccine safety monitoring system.

In Nigeria, only two studies within a Local Government Area (relatively small geographical coverage) in Lagos and Kaduna States examined health workers' reporting practices and attitudes [20, 21]. These studies relied on administered questionnaires (quantitative survey) to understand reporting practices and attitudes of health workers to reporting AEFI cases. These two studies have clearly shown the under-reporting of AEFI cases at Nigeria's primary health care (PHC) facilities [20, 21]. A study conducted in Jos demonstrated sub-optimal knowledge of AEFI among mother of children less than 23months who have identified AEFI but could not take appropriate actions following the AEFI [22]. Little has been known about the barriers to AEFI reporting from empirical quantitative studies that explored the knowledge, reporting practices, patterns and management of AEFI in Nigeria [20–23]. In this light, this study is the first to employ a qualitative approach for a robust and in-depth understanding of the barriers to optimal AEFI surveillance and documentation in Nigeria. Findings from this study will be helpful to vaccine manufacturers, regulatory agencies, donor agencies, national public health institutions, public health researchers, health development partners, including specialised United Nations Agencies, and Community Stakeholders, to improve detection, reporting and quality of documentation of AEFI that could guide causality assessment and restore confidence in immunisation services. It will guide the implementation strategy for AEFI surveillance

improvement, trigger a health system strengthening approach to PHC revitalisation and contribute to achievement of universal health coverage and, ultimately, the Sustainable Development Goals (SDGs).

## Methods

### Ethics statement

Ethical approval was obtained from the Kebbi State Health Research Ethics Committee, KSHREC, (under the Kebbi State Ministry of Health) with a registration number KSHREC: 106: 20/2021 in a letter dated 4th May 2021. Clearance was also obtained from the school of global health and bioethics through the faculty coordinator, Euclid University, before proceeding for the field investigation. The investigator obtained a formal verbal consent from the study participants.

### Study area

Kebbi State is a northwestern States of Nigeria with local and international borders. Locally, it borders Niger, Sokoto, and Zamfara States, while internationally, it borders Benin and Niger Republics [24]. Kebbi State enjoys a tropical climate characterised by annual rainfall ranging from 800 mm (northern part of the State) and 1000 mm (in the south), while temperature ranges from 21 to 40 degree Celsius [8]. However, climate change seems to have begun to impact, as seen in temperature excursions beyond 40 degrees Celsius (as much as 43 degrees) in recent times. There are 21 Local Government Areas (LGAs), 225 political wards and four traditional Emirates in the State [8]. It has a projected population of 5,119,659 (2021) and an under-one population of 204,786.

### Basic structure of AEFI surveillance system in Nigeria

At the health facility, the routine immunisation (RI) provider line-lists all the AEFI cases. In the event of a serious AEFI, the RI provider completes the line-list and AEFI reporting forms and transmits them to the LGA Disease Surveillance and Notification Officer (DSNO) and LGA Immunisation Officer or Routine Immunisation Officer (LIO/RIO) within 24 hours, using mobile phone or SMS or WhatsApp or any other telecommunication medium. The alert should also be sent to NAFDAC via SMS and it should trigger the use of an investigation form by the State Investigation Team. On the other hand, the monthly line-list of all AEFI cases is submitted to LGA DSNO. The summary record is also submitted to the LGA Monitoring & Evaluation (M & E) officer for data entry in the District Health Information System 2 (DHIS2) platform. If there is no detected case for the month, zero reporting can be done using the appropriate AEFI surveillance form.

There are two AEFI information/data management channels in place- Integrated Disease Surveillance and Response (IDSR) and District Health Information System 2 (DHIS2). The LGA DSNO captures the monthly summary of the AEFI report onto IDSR 003 and submits the line list to State DSNO while the LGA M & E officer captures the summary into the DHIS2 platform, which can be assessed by the individuals granted access at Zonal and National levels. The State DSNO captures the AEFI data into the AEFI database for proper archiving and analysis for decision-making. The State DSNO also transmits the AEFI data to the National level.

For those private medical institutions conducting immunisation services and treating AEFI cases, all AEFI reports are sent to their respective LIO/RIO within the institution or facility's catchment area, who will then share them immediately with the LGA DSNO. In the context of an immunisation campaign, vaccinators are to submit the report immediately to team

supervisors. The supervisors assess the validity of data without delay. The supervisors relay or submit the report to the Ward Focal Person, who is expected to report to the LIO, who in turn shares it immediately with the DSNO. Parents, caregivers, vaccinators, or healthcare workers within the communities can also notify the health facility about the AEFI cases, especially in a mass vaccination campaign setting, and this can be documented at the health facility for proper management.

## Study population

The study population includes the Routine Immunisation Service Providers at HF level, Routine Immunisation Officer, Local Immunisation Officers, Disease Surveillance and Notification Officers and Monitoring and Evaluation Office at the LGA level. At the State level, this included the State Director of Immunisation, and State Epidemiologist. State Partners were also interviewed, including WHO officers- Surveillance Focal Person, Routine Immunisation Officer, Data Assistant, and UNICEF's Communication for Development Officer. National Officers were also included in this study to enrich the findings and provide broad and upstream (national level) perspective on the AEFI surveillance system in Nigeria. Though not exhaustive, these are the key officers whose responsibilities have a strong bearing on immunisation service delivery, AEFI monitoring, reporting, and documentation and communication strategies. The study population is competent, qualified, and well-positioned to provide original and correct information regarding the barriers or challenges confronting optimal vaccine pharmacovigilance (safety monitoring) in Nigeria.

## Study design

A qualitative research design (using a ground theory approach) was adopted using in-depth interviews with key informants and focus group discussion. We conducted ten key informant interviews (IDIs) and two sessions of Focus Group Discussions (FGDs) among the study population to elicit information on the barriers impeding optimal AEFI surveillance and documentation in Nigeria.

## Sampling technique and data collection

Focus Group Discussions (FGDs) were conducted among the LGA-level Managers of the selected LGA Primary Health Care Department by the Principal Investigator (SAO). The selected health managers have job function with a bearing on immunisation service delivery, including AEFI surveillance. Conducting FGD sessions provided an in-depth understanding on the subject of investigation. We selected the three LGA-level Managers from each of the three distinct categories (by job functions) from the 21 LGAs. The selection of the three individuals from each category for the FGD was guided by performance based on the routine (secondary data) completeness data for AEFI by LGA in Kebbi State. Specifically, three officers from each distinct category (by job function) from three good performing LGAs and ditto for the poor performing LGAs. The selected LGAs by classification were Augie, Koko-Besse and Sakaba (good performing) and Aliero, Yauri and Zuru (poor performing) respectively. Consequently, a total of 18 officers from the three groups were selected for the FGD at the LGA level. The FGD was conducted in two sessions (a total of 18 participants, i.e., nine participants in each session).

The Principal Investigator (SAO) also conducted in-depth interviews of the key informants for senior officers at the State and National Levels. KIIs were conducted with the State Director of Immunisation, State Director of Public Health, and State Pharmacovigilance Desk Officer from NAFDAC, to provide an in-depth understanding of the phenomenon of interest, i.e.,

AEFI surveillance and documentation barriers and bottlenecks. At the federal level (national), the IDIs at the State level were complemented massively by interviews with senior officers-Directors from the pharmacovigilance and surveillance focal points at National Agency for Food and Drug Administration and Control (NAFDAC)and National Primary Health Care Development Agency (NPHCDA) respectively. Health Development Partners at the State level, such as the WHO officers- RI and Surveillance Focal Points, and Data Support Officer and Communication for Development Officer for UNICEF were also interviewed.

Overall, a total of 10 IDIs and 2 FGD sessions (involving 18 participants in the two sessions) were conducted for this study. There was no case of refusal to participate in this study. The IDIs and FGD sessions were either virtual or physical, depending on the convenience and availability of digital technology on both sides (interviewer and the study participants). Most of the interviews were conducted face-to-face with the study participants. These interviews were either held in the office or at end of a joint programme review meeting involving LGA Level Managers. The interviews were recorded in audio and stored in an archive. A few (two) virtual interviews (for the key informants) were conducted with national level officers from the NPHCDA and NAFDAC. The qualitative interview for each FGD session lasted for a maximum of an hour, while the IDI lasted for a maximum of 50minutes. The participants were provided with their interview transcriptions for validation before data analysis. Ultimately, the decision to discontinue recruitment into the study was guided by data saturation when no new information was generated from the participants' interviews.

## Interview guide

Though the participants were given ample opportunities to freely express their views and opinions about the subject of interest, an interview guide serves as an aid for the interviewer (principal investigator) to elicit answers to the research questions. The interview guide was constructed from the research objectives and the existing literature on the subject of investigation.

## Data analysis

The researchers cleaned the data, processed and analyzed the data manually for the qualitative survey using a thematic framework approach. The process involved verbatim transcriptions of returned audios from professional transcribers who were not part of the investigation. The transcripts were screened, edited, and double-checked independently by the research team for accuracy, consistency, and clarity. The note taken during the interview sessions were used to guide the screening and editing to improve the overall quality of the transcription (transcribed data). Further analyses involved sorting, coding, and thematic identification. The coding was developed based on the objectives of the study, the research tools, and the responses elicited from the study participants.

We generated themes and sub-themes that were consistent and relevant which provided a framework for a better understanding of the research topics and objectives. Subsequently, we used the content analysis method to generate qualitative data. Content analysis compresses texts into replicable, fewer content categories based on clear coding rules [25]. Therefore, I (SAO) summarised responses to each theme and sub-theme and ensured that relevant and important expressions like quotations by the respondents were reported verbatim to enrich the outputs of the study.

## Trustworthiness of the qualitative study

This qualitative research study is trustworthy based on these criteria: credibility, transferability, dependability and confirmability.

**Credibility.**    The principal investigator had sustained engagements with key stakeholders in AEFI surveillance and Immunization space within the study setting through IDI and FGD sessions. The study research questions were based on a decade of professional work experience as a development partner and routine AEFI data collection (secondary data). Hence, the likelihood of misinterpretation and misrepresentation of findings and data is almost nil. Further, the findings had undergone peer review within the faculty to also ascertain the credibility of the process and findings.

**Transferability.**    This refers to the generalizability of the inquiry. The methods of investigation were clear, and the categories and number of participants and bases for the purposive selection were detailed in the manuscript. This is plausible in settings with similar structures and reporting systems. The findings are also transferable given the similarities in the underlying health system factors in-country and the national stakeholders' perspectives. These elements largely confer the generalisability of the study findings.

**Dependability.**    The author made an effort to achieve dependability by ensuring that the research process is clear, replicable and well-documented. The recordings were provided to the review and editorial teams for transparency and potential audit purposes.

**Confirmability.**    It is established when the researcher can demonstrate credibility, dependability and transferability. Essentially, it has to do with demonstrating how the researchers' interpretation and findings are derived from the data. Hence, the authors have clearly demonstrated confirmability in this study.

## Research team

Regarding the research team, the principal investigator (SAO) managed the sessions effectively, given that the researcher was orientated in the past and had moderated a few FGD sessions for the WHO Academy in late 2020. He has also published two studies using the qualitative research approach. The other authors (MB and LCK) have either had training or had published qualitative studies. In addition, SAO (male) is a medical doctor, works within the disease surveillance and immunisation sphere with the World Health Organization and a doctoral student at the time of the research. He has extensively worked in the study setting and developed excellent professional relationships with the key stakeholders, especially at the primary health care level. LCK (male) is a senior author, had a doctoral degree, had investigated and published on vaccine hesitancy using a qualitative approach. He is the Faculty Coordinator at the School of Global Health and Bioethics, Euclid University. The third author (male), MB had a doctoral degree in Biostatistics and Statistical Science and works with the Euclid University.

## Results

We interviewed 10 key informants and conducted two FGD sessions (involving 18 participants). The participants in this study were predominantly males. Indeed, only two women participated in this study reflecting the gender imbalance in the health workforce in the study setting. The median age of the FGD participants was 44years with the youngest being 33years while the oldest was 52years. For the length of years in service, the median years of service was 23.5years and the highest and lowest years in service were 34 and 11years respectively (see Table 1 for the demographic characteristics of the participants in this study).

Asides the brief quantitative description of the study participants, this section also highlighted the results from qualitative analyses conducted, including Table 2 that showed some key results (findings) from the interviews and FGD sessions conducted.

**Table 1. Basic characteristics of participants at the FGD sessions.**

| S/N | Name | Unit | Sex | Age | LGA | Qualification | Service Years |
|---|---|---|---|---|---|---|---|
| 1 | SB | Surveillance | M | 43 | ZURU | OND (EH) | 15 |
| 2 | MUR | Immunisation | F | 45 | ZURU | BSC | 27 |
| 3 | AD | M and E | M | 45 | ZURU | OND (HIR) | 24 |
| 4 | KGP | Surveillance | M | 48 | SAKABA | OND (CHEW) | 25 |
| 5 | MG | Immunisation | M | 44 | SAKABA | OND (CHEW) | 27 |
| 6 | UHR | M and E | M | 47 | SAKABA | OND (EH) | 23 |
| 7 | AJ | Surveillance | M | 59 | AUGIE | OND (CHEW) | 34 |
| 8 | SS | Immunisation | M | 52 | AUGIE | OND (EH) | 28 |
| 9 | YMK | M and E | M | 44 | AUGIE | OND (EH) | 20 |
| 10 | AS | M and E | M | 55 | ALIERO | HND | 25 |
| 11 | LA | Immunisation | M | 40 | ALIERO | OND | 22 |
| 12 | UA | Surveillance | M | 41 | ALIERO | OND | 13 |
| 13 | FA | M and E | M | 33 | KOKO BESSE | OND (EH) | 11 |
| 14 | SA | Surveillance | M | 43 | KOKO BESSE | OND (CHEW) | 23 |
| 15 | YA | Immunisation | M | 40 | KOKO BESSE | OND (EH) | 12 |
| 16 | MA | Immunisation | M | 42 | YAURI | OND (CHEW) | 20 |
| 17 | MG | M and E | M | 47 | YAURI | OND (CHEW) | 25 |
| 18 | KM | Surveillance | M | 41 | YAURI | OND (CHEW) | 24 |

**Basic Characteristics of Participants for IDI**

| S/N | Name | Designation | Sex | Age | Level | Qualification | Service Years |
|---|---|---|---|---|---|---|---|
| 1. | MTR | Surveillance | M | 40 | State | MPH | 13 |
| 2. | LS | M and E | M | 29 | State | HND | 7 |
| 3. | KY | Surveillance | M | 55 | National | PhD | 30 |
| 4. | UE | Pharmacovigilance | F | 50 | National | MSc | 22 |
| 5. | SSA | Communications | M | 40 | State | BSc | 10 |
| 6. | AD | Immunisation | M | 39 | State | MPH | 10 |
| 7. | AK | Immunisation/ Disease Control | M | 52 | State | MPH | 34 |
| 8. | AB | Surveillance/Epi | M | 52 | State | MPH | 32 |
| 9. | GN | Pharmacovigilance | M | 35 | State | BSc | 7 |
| 10. | JU | Surveillance/ Immunisation | M | 33 | LGA | HND | 10 |

## Theme: Bottlenecks and challenges confronting AEFI surveillance system

**Sub-theme: Impediments to optimal AEFI surveillance and documentation.** *Human resource-related issues.* From the in-depth interviews and the focus group discussion sessions with the LGA level officers, State and National Level officers, there was a preponderance of assertions about poor knowledge of health workers on what needs to be reported, including the case definition and the reporting channel, the documentation procedures, and the value of reporting serious and non-serious AEFI cases. These above-mentioned elements emerged as barriers/impediments to effective AEFI surveillance and documentation. However, the limited training conducted on AEFI was essentially targeted at health workers rendering RI service, whereas all health workers should be trained on AEFI.

A flicker of evidence from the FGD suggests poor motivation as an impediment. Health workers are not incentivised to report and document AEFI. This is connected to the poor attitude to AEFI reporting. The fear of reprimand/punishment or fear of being under-rated/blamed was mentioned during the interview. From the conversation, health workers do not want to report to avoid being punished or blamed as it questions their professional competence.

**Table 2. Summary of some key results from the interviews and FGD conducted in this study.**

| Themes | Sub-themes | Responses |
|---|---|---|
| Human resources-related issues | Knowledge of what needs to be reported/knowledge of case definition, documentation and channel of reporting | "...the service providers are sometimes not well knowledgeable about the case definition or even what to report even if these cases are detected"-**WHO LGAF (IDI)**<br>"...some of the health workers don't know the value of this AEFI that's why they are not filling these AEFI forms"- **NAFDAC Kebbi (IDI)**<br>"There is a knowledge gap in some of the health workers because only a few have an in-depth knowledge of what AEFI is and how to report it and the appropriate channels"- **LGA-level POs, Batch B (FGD)** |
| | Fear of reprimand/punishment/ consequences/ feeling of professional incompetence | "...the reasons people are not reporting mostly is sometimes poor knowledge of what needs to be reported, and the fear of reprimanding- feeling that the cause of the AEFI is actually a result of inadequacies in their professional practice, which is wrong." **NAFDAC NPVFP (IDI)**<br>"The health workers do not want to report AEFI thinking that they will be blamed for the adverse event..."—**LGA-level POs, Batch B (FGD)**<br>"...they are afraid of reporting AEFI thinking that they will be punished or be under-rated that they are not qualified or patient enough to give vaccinations"- **LGA-level POs, Batch B (FGD)** |
| | No focal person for AEFI surveillance | And then, there are no specific responsible officers at the LGA level, at the State level, there is no focal person that is responsible for collation or analysing AEFI data" -**WHO LGAF (IDI)**<br>"...right from the state level (government), there is no person or personnel in charge of this AEFI, but there should be a focal person at the LGA level and the state level to monitor/supervise this RI AEFI documentation and surveillance"—**LGA-level POs, Batch B (FGD)** |
| | Heavy workload/poor motivation | "...multiple data tools and only one person or RI provider is required to fill or handle all these data tools"- **LGA-level POs, Batch B** (FGD<br>"...heavy workload in the facilities and poor motivation"- **LGA-level POs, Batch A (FGD)** |
| | Weak/lack of supportive supervision | "From my experience, there is a lack of qualitative supervision, especially on that AEFI surveillance and documentation"- **LGA-level POs, Batch B (FGD)**<br>"...the level of documentation in some of the health facilities is very poor, especially in the rural areas, where they know that they don't receive visitors on a routine basis, so they tend to be idle in terms of proper documentation of these AEFI cases" -**WHO SDA (IDI)** |
| | Commitment, mindset, and attitude | "..the other thing apart from funding is the mindset of our people (health workers), their level of commitment is sub-optimal, and there is a low level of education of our health workers at the lower level, lack of proper data archiving system even in our LGAs"- **DoI, SPHCDA- Kebbi**<br>"...societies (communities) are not the problem. Our problems are our counterparts at the PHC level. There is no commitment from the RI providers"–**UNICEF Comm FP (IDI)** |
| Caregiver/Parental Factor | Information deficit, low awareness; poor feedback | "...but most of the people don't have this information, even if it has happened, they won't know they are supposed to bring the child to the health center"- **LGA-level POs, Batch B (FGD)**<br>"...and feedback is very poor because they will think it is not all that important to report AEFI to the health facility"- **LGA-level PO, Batch A (FGD)** |
| Governance and Leadership | Weak coordination meeting/moribund AEFI committee; less priority on AEFI surveillance by the government | "...we tend to have the AEFI committee within the LGA and the State, who are actually supposed to take responsibilities of monitoring and investigating serious AEFI. Because there is inadequate logistics provision for them to hold regular meetings, carry out their functions as may be required, these meetings are barely done within LGA and State, except during campaigns because funds are provided during campaigns"—**NAFDAC NPVFP (IDI)**<br>"...the government places less priority on the AEFI compared to disease surveillance"- **NAFDAC Kebbi (IDI)** |

(*Continued*)

**Table 2.** (Continued)

| Themes | Sub-themes | Responses |
|---|---|---|
| Funding and Logistics | No dedicated resources for AEFI; lack of AEFI kit (emergency medications); data tools shortage | *"...the state government is not helping even financially to improve the reporting, detection, and documentation and use of data for action"*–**DoI, SPCHDA-Kebbi (IDI)**<br>*"From the HF to LGA and State level, there is no logistics arrangement, and in terms of the timeline, it is a bit challenging for the DSNOs to report what they have as at when due when they don't data tools at their level. And at the state level, if they have a signal or cluster for them to go and investigate and then report to the next level, it is a challenge for them because there are no dedicated resources for investigating AEFI at that level"*- **WHO SFP (IDI)**<br>*"Most of the health facilities lack emergency drugs to manage the AEFI, especially the serious cases..."*—**LGA level POs, Batch B (FGD)**<br>*"Lack of enough data tools to report those cases or mismanagement of those data at the health facilities..."*–**LGA-level POs, Batch B (FGD** |
| Socioeconomic factors and infrastructural deficits | Distant to health facilities; bad terrain; mode of transport and poverty | *"Sometimes the distance from the community to the health facility is too far, and sometimes you find out that there is only one health facility. The bad terrain in some of our communities, mode of transportation, and poverty make it difficult for the caregivers to return to the health facility to report AEFI"*- **LGA-Level POs, Batch A (FGD)** |
| Insecurity | Threat to health workers' lives- banditry and kidnapping | *"Security issue is another challenge, especially for those LGAs in the southern part, there is a risk of insecurity- bandit attack and so on"*- **WHO SDA (IDI)**<br>*"Another challenge is insecurity-fear of being killed by bandits"*- **LGA-level PO, Batch A (FGD)** |
| Ineffective communication and information management | health education paucity; poor feedback from NEC; satisfaction with immunisation services | *"...lack of enough advice to the caregivers from the health workers to bring their child to the health center if he/she has any effect/signs after immunisation"*–**LGA-Level POs, Batch B (FGD)**<br>*"...the system works partially because most of the time, most of the reports that we sent do not have feedback, without feedback you will not have the confidence to continue reports, that is why the reporting is low"*- **LGA-level POs, Batch B (FGD)**<br>*"...no knowledge about AEFI and RI service providers will only vaccinate without giving sensitization on AEFI, and feedback is very poor because they will think it is not all that important to report AEFI to the health facility"*- **LGA- level POs Batch A (FGD)**<br>*"It is not easy to get feedback very quickly to tell the caregivers so that they will feel satisfied with the immunisation services"*–**LGA-level POs, Batch B (FGD)** |

A few interviewees (IDIs) also pointed out the health workers' poor or lack of commitment as an impediment to optimal surveillance and documentation. Also, from the discussions and interviews, a lack of designated officers for AEFI surveillance and documentation was pointed out as an impediment. This is linked to the heavy or excessive workload mentioned by interviewees- the same officers conducting RI at the HFs are expected to detect, report and document AEFI, among other responsibilities. No specific desk officer for AEFI at the State and LGA level. Supportive supervision is lacking or inadequate at the health facility level. Visits by senior supervisors are hampered by other systemic issues.

*"...the reasons people are not reporting mostly is sometimes poor knowledge of what needs to be reported, and the fear of reprimanding- feeling that the cause of the AEFI is actually a result of inadequacies in their professional practice, which is wrong."* **NAFDAC NPVFP (IDI)**

*"...the service providers are sometimes not well knowledgeable about the case definition or even what to report even if these cases are detected. And then, there are no specific responsible officers at the LGA level, at the State level, there is no focal person that is responsible for collation or analyzing AEFI data"* -**WHO LGAF (IDI)**

*". . ..Only the RI providers are invited to training, while the rest of the health workers are not trained and not knowledgeable about AEFI surveillance and documentation. At least all the health workers should know how to fill the form/report properly"*- **LGA-level POs, Batch B (FGD)**

*". . .societies (communities) are not the problem. Our problems are our counterparts at the PHC level. There is no commitment from the RI providers"*–**UNICEF Comm FP (IDI)**

*". . .the other thing apart from funding is the mindset of our people (health workers), their level of commitment is sub-optimal, and there is a low level of education of our health workers at the lower level, lack of proper data archiving system even in our LGAs"*- **DoI, SPHCDA-Kebbi**

*". . .some of the health workers don't know the value of this AEFI that's why they are not filling these AEFI forms"*- **NAFDAC Kebbi (IDI)**

*". . .heavy workload in the facilities and poor motivation"*- **LGA-level POs, Batch A (FGD)**

*". . .multiple data tools and only one person or RI provider is required to fill or handle all these data tools"*- **LGA-level POs, Batch B** (FGD)

*". . .the level of documentation in some of the health facilities is very poor, especially in the rural areas, where they know that they don't receive visitors on a routine basis, so they tend to be idle in terms of proper documentation of these AEFI cases"* -**WHO SDA (IDI)**

*". . .right from the state level (government), there is no person or personnel in charge of this AEFI, but there should be a focal person at the LGA level and the state level to monitor/supervise this RI AEFI documentation and surveillance"*—**LGA-level POs, Batch B** (FGD)

*"From my experience, there is a lack of qualitative supervision, especially on that AEFI surveillance and documentation"*- **LGA-level POs, Batch B** (FGD)

*Caregivers' and parental factors.* On the part of the parents/caregivers, evidence from the FGD with the LGA level officers clearly indicates a low level of awareness to report AEFI cases. This is linked to the poor sensitisation or information deficit about AEFI during health education prior to immunisation session (bearing in mind the education profile of women in the study, especially in the rural communities).

*". . .and feedback is very poor because they will think it is not all that important to report AEFI to the health facility"*- **LGA-level PO, Batch A (FGD)**

*". . .but most of the people don't have this information, even if it has happened, they won't know they are supposed to bring the child to the health center"*- **LGA-level POs, Batch B (FGD)**

*Governance and leadership.* Evidence from the IDIs and FGD sessions suggests that the AEFI committees at LGA and State levels are moribund (docile)- no coordination meeting, no supervisory visit and investigation done. Most of the policies as enshrined in the national guideline are not implemented, which could be explained by the lack of dedicated budget (funding) to AEFI surveillance activities, poor commitment, and non-prioritisation (lack of interest as defined by some interviewees) of AEFI surveillance and documentation compared to diseases surveillance.

*". . .from detection, reporting and the use of data for action, there are really a lot of challenges. It is only WHO that is passionate about it; the state government is not helping even financially*

*to improve the reporting, detection, and documentation and use of data for action"*- **DoI, SPHCDA-Kebbi (IDI)**

*"Lack of interest by the government to support electronic data management system for AEFI"*- **DoI, SPCHDA-Kebbi (IDI)**

*Funding and logistics*. From the IDI and FGD sessions, it is incontrovertible that the AEFI surveillance system in Nigeria lacks the much-needed funding to support the transport of personnel and commodities at the health facility level, to support AEFI committees' coordination meetings, supervisory visits, and investigation of serious cases. Based on the information provided, data tools for reporting and documentation are not routinely available except during SIAs/OBR. Similarly, most health facilities lack AEFI kits to provide emergency treatments for serious and non-serious AEFI, and referral mechanisms are not in place.

*"Then, issues that also have to do with logistics, for example, if somebody comes to the health facility, get immunised, and the person is from a hard-to-reach area (HTR) of the LGA, the health care workers will need some logistics support for adequate reporting and investigation. Once the logistics support is not available for the health care workers to do that work, it will never be done"*- **NAFDAC NPVFP**

*"The DSNO is talking about the case of hard-to-reach areas when the service providers conduct outreach service before they go back for another session- maybe they will go back a month later to that settlement. The quality-timeliness of reported cases has been lost"*- **LGA-level POs, Batch A (FGD)**

*". . .none of the HF has AEFI kits"*—**LGA-level POs, Batch A (FGD)**

*Socio-economic factors and infrastructural deficits*. These are essentially factors operating outside of the health service delivery system but impact the health system's capacity to provide service to the population in need. For example, poverty at the individual and population level affects the ability of parents and the community to access health services, including reporting and managing AEFI cases. Many of the rural villages in the rural areas have bad terrains, making it difficult to access health care. Even surveillance and immunisation outreaches are hampered, as health workers find it difficult to access such locations. Geographical barriers (far distances and challenging terrains) to health services are impediments in this environment, especially in rural areas.

*"Sometimes the distance from the community to the health facility is too far, and sometimes you find out that there is only one health facility. The bad terrain in some of our communities, mode of transportation, and poverty make it difficult for the caregivers to return to the health facility to report AEFI"*- **LGA-Level POs, Batch A (FGD)**

*Insecurity*. A contemporary issue that emanated from the qualitative interviews was the role of insecurity. The interviewees indicated that fear of being killed by armed bandits and kidnapping impede AEFI surveillance activities by limiting access of both the health workers and recipients of health services (clients, caregivers and parents). Health workers have been victims of such attacks.

*"Security issue is another challenge, especially for those LGAs in the southern part, there is a risk of insecurity- bandit attack and so on"*- **WHO SDA (IDI)**

*"Another challenge is insecurity-fear of being killed by bandits"*- **LGA-level PO, Batch A (FGD)**

*Ineffective communication and information management.* The negative role of poor communication and information management by the operators (local and national) came to the glare during the two FGD sessions. It impacts the knowledge of the caregivers and their capacity to report AEFI. Similarly, the consistent lack of feedback from the national level on the reported cases was a disincentive to continued AEFI reporting, as indicated in the discussions. The quality of sensitisation (health education session) and engagement with the parents were poor, bringing dissatisfaction and loss of confidence in the immunisation services. No evidence suggests community engagement and media publicity to raise awareness and stimulate AEFI reports at the population level.

*"The DSNO is talking about the case of hard-to-reach areas when the service providers conduct outreach service before they go back for another session- maybe they will go back a month later to that settlement. The quality in terms of timeliness of reported cases has been lost"*- **LGA-level POs, Batch A (FGD)**

*"The attitude of some health workers towards caregivers is not good, and the information given to the illiterate caregivers in the rural areas is not sufficient to encourage immunisation and AEFI reporting. And usually, there is no feedback, even when few cases are reported and investigated. . ."*- **LGA-level POs, Batch B (FGD)**

## Discussion

### Impediments to optimal AEFI surveillance and documentation in Nigeria

Regarding the bottlenecks and challenges confronting optimal routine AEFI monitoring, reporting and documentation in Nigeria, the respondents working at the health facilities provided important details, such as the work overload (arising from human resource deficit), lack of knowledge about the value or relevance of the reporting and documenting AEFI cases (serious and non-serious) and sub-optimal understanding of the reporting format and cumbersome documentation process. Similarly, evidence from higher-level stakeholders' interviews showed poor knowledge of what needs to be reported.

In a study conducted in Alimosho LGA, Ogunyemi and colleagues reported not knowing about the notification and reporting system and process, inability to find forms, and not considering the event as related to immunisation are some of the barriers to optimal AEFI reporting [21]. In the same vein, a study conducted at Sabon Gari LGA, Zaria, in the northern part of Nigeria, reported not considering the event related to immunisation, ignorance of the reporting processes, and time constraints, perhaps due to competing demands or excessive workload [20]. Similar to this study's finding, Yamoah et al. reported from a study among health care professionals in Ghana that cumbersome bureaucratic process of reporting, complex data collection tools and busy work schedule (work overload) were impediments to reporting and documentation of AEFI [26]. Another study from Ghana, published in 2020, highlighted multiple barriers to reporting which were largely similar to this and previous studies [27]. The identified barriers were lack of knowledge or training and not believing an AEFI was serious enough to be reported [27]. Unlike many previous studies, including this study, reporting forms were readily available at the health facilities at the time of interviews in most health facilities [27].

A study from Zimbabwe that analysed spontaneous reporting, Vigibase, from 1997 to 2007 also documented competing work priorities, insufficient reporting resources and inadequate

knowledge deficit due to insufficient training of health workers [28]. Similarly, Malande et al. reported from Kenya that excessive paperwork and effort and the non-availability of reporting tools for documentation were impediments to AEFI reporting and documentation [29]. A study from Gaza in the Middle East identified some impediments to AEFI reporting and documentation, such as high workload, not considering AEFI related to immunisation and absence of guidelines, protocols and notification forms [30]. Similarly, a study from the Far East, India, among Pediatricians in Kerala reported, among others, difficulties in reporting processes (paper-based reporting forms) and time constraints and work pressure, less formal training (comparatively lower knowledge of reporting and reporting processes) among private health care providers as barriers to AEFI reporting [31].

Findings from studies from developed countries such as Australia, Canada, United States of America buttressed findings from this work and other published work conducted in developing countries. For example, studies conducted in Australia, the United States of America (USA), the United Kingdom, and Canada indicated that unclear definition of reportable AEFI, not knowing AEFI reporting was expected or events deemed not serious enough to be reported were barriers to AEFI reporting and documentation [32–34]. Like other studies from some developing countries cited above, two studies from the USA also reported that health care professionals mentioned a lack of awareness of the reporting procedure or confusion as to who is responsible and when to report as barriers to optimal AEFI reporting and documentation [32, 34, 35]. Lastly, time constraint/pressure to complete the reporting forms was identified as a barrier by studies conducted in Australia and the United Kingdom [33, 34].

A global review of records and immunisation safety monitoring data between 2010 and 2019 found gaps in global immunisation safety monitoring, especially in Africa, Western Pacific, European and Eastern Mediterranean regions [17]. The report also identified a lack of reporting tools and poor health workers' understanding of AEFI [17].

This study also found an attitudinal problem of the health workers as a significant impediment as commitment is lacking in many cases. A study conducted at Sabon Gari LGA, Kaduna State, northwestern Nigeria, among primary health care workers identified two attitudinal elements- forgetfulness and lack of interest- as barriers to AEFI reporting [20]. As reported by the health workers, forgetfulness may be related to excessive workload and non-prioritisation of AEFI reporting and documentation as an integral part of immunisation programme. The apathy could be explained by a poor understanding of the benefit and integral nature of AEFI in immunisation programmes. It could also stem from demotivation and poor working conditions of health workers in the study setting. Internationally, Mehmeti et al. reported the lack of interest as a barrier to reporting AEFI in Albania [36].

Fear of reprimand, i.e., feeling that the cause of the AEFI is a result of inadequacies in their professional practices, remains an important impediment to optimal reporting and documentation of AEFI cases at the PHC level. Several published papers from within and outside Nigeria have identified fear of reprimand, fear of personal consequences/punishment, fear of being victimised, or fear of raising public alarm as part of the major barriers to reporting and documentation of AEFI [17, 27, 37–41]. Unlike a few other studies, this piece of work did not find that the fear of litigation or adverse events was not reported because of inherent trust that licensed vaccines are safe [32, 42, 43]. The preponderance of the fear of reprimand or punishment and fear of victimisation or personal consequences as a barrier to detection, reporting and documentation of AEFI (frequent findings in LMICs) are all a reflection of poor or lack of training on AEFI (pre- and in-service training), weak supportive supervision, non-prioritisation of AEFI and weak governance. As mentioned above, the poor situation requires urgent attention by stakeholders at all levels to address the background causes or triggers of the identified factor/barrier. Therefore, this work adds to the growing international call for introducing

courses on AEFI -causes, management, surveillance and reporting, to medical and nursing educational curricula. Previously published works have also confirmed this increasing need, as observed in Canada, the USA, Europe and Australia [32, 33, 44–46]. Beyond the pre-service training in higher health institutions (medical and nursing schools and college of health technology for community health workers), there should be periodic statutory in-service training for immunisation service providers and surveillance officers. These, among other measures, will reverse the current sub-optimal performance of AEFI surveillance and documentation in LMICs (including Nigeria) and beyond.

Concerning the role of the caregiver's educational level and adherence to immunisation services, the higher the level of educational status of the mothers/caregivers, the more likelihood of adherence to the immunisation schedule [47] By extension, this role of mothers' educational status could impact the likelihood of detection and reporting of AEFI by caregivers. In our study setting, the educational status of women is lower than men, where the majority live in the rural environments. These and other socio-ecological elements interacting together might have negatively influenced their capacity to use the limited health information provided by the health workers. This could explain the low level of awareness and sub-optimal AEFI reporting by parents/caregivers in this study.

The government's non-prioritisation of AEFI surveillance and documentation as a core component of routine immunisation service delivery was identified as an impediment to optimal AEFI surveillance and documentation in this study. This also translated into the lack of funding, weak stewardship and governance of pharmacovigilance in Nigeria. It has also manifested in the sub-optimal coordination between the national regulatory agency and immunisation programme (EPI) implementing agency, as seen in the slow pace of efforts towards implementing electronic AEFI surveillance, specifically the Med Safety App, being promoted by NAFDAC- the national regulatory agency. Weak coordination and stewardship are commonly reported in LMICs [17, 48]. Further, this study indicated that partner agencies, such as WHO, have been the vanguard of strengthening the AEFI surveillance system in Nigeria. Few reports and reviews have identified weak political will and coordination by governments in LMICs as part of the barriers to a strong vaccine safety monitoring system, i.e., AEFI surveillance and documentation [38, 49].

A sequela to the above factor is the non-availability of a dedicated budget line or funding stream to support routine AEFI surveillance and documentation activities (except during campaigns- SIA or OBR). During SIA or outbreak response, partners agencies, donors and government often pool resources together to train health workers, provide logistics and set up a vaccine safety monitoring system during the campaign implementation. This study also found that State and LGA AEFI committees are moribund (outside the campaign and OBR modes) due to the non-availability of logistics to hold regular meetings and undertake monitoring and investigation of serious AEFI cases. Evidence from global studies and expert reviews indicated little or no funding dedicated to supporting vaccine safety surveillance and pharmacovigilance in LMICs, especially in Africa [17, 49–51]. Thus, corroborating this study's finding.

The funding and logistics challenges limit supportive supervision of AEFI surveillance activities, especially in rural areas. This was clearly pointed out by critical stakeholders that were interviewed in this study. Supportive supervision could be defined as a range of measures to ensure that personnel carry out their activities effectively through direct, regular personal contact to guide, support and assist designated staff to improve their work competency [52]. It is one of the operational components of the "Reaching Every District" strategy, aimed at improving immunisation performance in every district (including health facilities), especially in the African region, where access and utilisation of immunisation services are suboptimal [53]. Supportive supervision has improved equity and immunisation service delivery efficiency

[52]. A study conducted by Gbenewei et al. reported that the deployment of supervisors during the campaign (mass immunisation programme) might play an important role in improving the identification and reporting of suspected AEFI [54]. Further, a randomised control trial in Cameroon to assess the effect of sending weekly SMS and weekly supervisory visits on AEFI reporting rate during a meningitis immunisation campaign conducted in Cameroun using Meningitis A Conjugate vaccine, found supervision more effective than SMS or routine surveillance in reporting AEFI [55].

However, a pointer to the complementary roles of adequate funding and logistics in improving and achieving quality immunisation services (AEFI surveillance and documentation inclusive) has been established in the literature, corroborating this study's funding. For example, 'a quasi-experimental study' conducted in India, a lower-middle-income country, to assess the effect of supportive supervision on immunisation session practices indicated that there is a need to address systemic issues, such as essential logistics and supply chain management and financial resources that could complement the supportive supervision strategy to improve immunisation service delivery [56]. Similarly, a review of immunisation programmes conducted across 20 East and Southern African countries from 2012 to 2018 found that half of the countries examined had inadequate operational funding, evidence of cancellation of outreach services and supportive supervisory visits due to lack of funding or operational funding, while nine countries had weak vaccine stock management system characterised by stock-out of vaccines and supplies [57]. Indeed, the findings in this study have been strongly buttressed by reports across similar settings in LMICs, indicating the identified challenges of human resource deficits, inadequate logistics and vaccine management system, lack or inadequate operational funding for transport and fuel for cold chain equipment, among others [58, 59]. From the above, there is a need to invest in the operational readiness (fueling, operational cost e.t.c), vaccine management and supply chain system, and of course, specific attention should be paid to AEFI supportive supervision being aided by the aforementioned functional systems in place.

The study also found a lack of prepositioned AEFI kits to manage serious and non-serious AEFI cases at health facilities as one of the challenges of achieving a functional and responsive AEFI surveillance system. Only a report from India mentioned a similar challenge with the availability of AEFI kits at the HFs, where vaccination services were provided [60]. There was no dedicated transportation facility or mechanism to facilitate referral mechanisms for serious AEFI cases in the study environment. The national policy stipulates that treatment of AEFI is free, but this is not the case in practice, as caregivers buy drugs out-of-pocket to treat such adverse event/reaction(s). Similarly, there was no ancillary support, such as psychosocial support services, to address the mental and psychological impacts of the adverse events/reactions.

Further, a lack of feedback and/or non-recognition of their contributions was a disincentive to reporting AEFI at the health facility level. A published study from western Europe indicated non-existent or inappropriate feedback [61], identical to the finding in this study. Similarly, the study from Albania, Eastern Europe, indicated that most of the health workers never got feedback- a lack of communication from the vaccine safety stakeholders [36]. Masuka et al. acknowledged the importance of appropriate and timely feedback to reporters of AEFI to increase the number and quality of AEFI reports [28]. Even in developed countries like Canada, a study among family physicians indicated that they reported uncertainty about causality between the vaccine and the event [32], suggesting a lack of feedback and possibly an information or knowledge deficit. The study also reported the need to give physicians feedback and avail them of learning opportunities on vaccine-associated event reporting [32].

The timeliness and quality of feedback on reported AEFI determine its usefulness to the regulators, vaccine manufacturers, reporters, and the vaccinees or the parents of the vaccinees

and the public [61]. A lack of or inappropriate feedback to health workers and parents or vaccinee is a considerable disincentive to further or future reporting as interests tend to wane [51, 62, 63]. Ultimately, confidence in the immunisation programme gets eroded.

Another emerging threat to AEFI surveillance activities at the PHC level is the growing insecurity in northwestern Nigeria. This impacts the frequency of supportive supervision, data transmission, investigation of reported serious AEFI cases and disruption of health services. No investigator has previously reported the role of insecurity or conflict on this subject. Insecurity impacts access and quality of health service delivery. Three relevant pathways were identified based on a study from the Democratic Republic of Congo (DRC), namely violence, mobility restriction and resource availability [64]. The pathways manifest in the availability and quality of the health workforce, access to medicine, health commodities and medical technology, health workers' and client access to health facilities or restriction of field investigation, transport and analyses of biological samples, etc. All these elements impact the capacity of the conventional health system, including surveillance, to rapidly detect, report and investigation AEFI cases. Therefore, adaptive strategies are highly warranted in the context of insecurity, as seen in Nigeria and countries like DRC and Somalia, to mitigate the disruption of health services, including the surveillance systems.

Some relevant measures or strategies could be employed to ensure AEFI surveillance is uninterrupted in the context of insecurity. Such strategies could emanate from the lesson learned as the polio program evolved in Northeastern Nigeria, where Boko Haram Insurgency turned the region into the last WPV sanctuary in Nigeria and Africa. Similarly, armed banditry, sporadic violence and kidnapping have escalated in the northwestern region- this study's setting, and sometimes health workers are the victims [65]. Some of the strategies implemented in security-compromised setting were deployment of community informants to detect and notify AFP cases from inaccessible or partially accessible areas, the use of mobile phone applications for Auto-visual AFP Detection and Reporting (AVADAR) by health workers and informants in challenging terrains and insecure areas, civilian-military security partnership and health service delivery among others [66–69]. These strategies can be applied to sustain or build resilience at the PHC level for AEFI surveillance and documentation. The Med Safety App on the android phone being piloted to promote self-reporting of AEFI is akin to the AVADAR innovation that has proven effective in complementing conventional AFP surveillance, particularly in security and geographically challenged locations.

This study's finding also indicated inadequate communication between the RI providers and the clients (caregivers/mothers). Adequate health education services are not often provided before vaccine administration, such that caregivers/mothers will be able to make well-informed decisions regarding AEFI detection, reporting and follow-up actions. The paucity of information provided by health workers, especially in a setting where female education remains low, could contribute to high drop-out and the high number of partially immunised. This may be largely due to a poor understanding of AEFI and the benefits of full immunisation for eligible children. Oku et al. pointed out that the quality of information provided to caregivers in rural health facilities in Bauchi State, Northeastern Nigeria, focused on polio and were not detailed, perhaps due to a shortage of health personnel [70]. In the Guruve district of Zimbabwe, Constantine and colleagues found that caregivers were not reporting AEFI to the health facilities because they were not educated enough to know that they have to report the events [37]. These findings buttressed our observation on the communication gap between RI service providers and parents/caregivers, impacting immunisation service delivery, including AEFI surveillance and documentation.

Poor community awareness and sub-optimal community linkage were clearly identified as one of the weaknesses of Nigeria's current AEFI surveillance system. The awareness level and

understanding of the need to report AEFI is low in this environment. Inherently, the EPI in Nigeria remains challenged by sub-optimal awareness and understanding of the importance of routine immunisation or vaccination services (SIA/OBR inclusive). Studies have shown that a low level of awareness and weak community linkage impedes the optimal uptake of routine immunisation services and other PHC services [71–75]. In the same vein, low awareness, poor community linkage, and engagement contribute to weak pharmacovigilance, especially low reporting, poor documentation, and limited vaccine and drug safety information [17, 49, 51]. Hence, there should be a systematic engagement of community informants supported with the necessary tool to intensify AEFI detection and reporting from the community level. Media publicity and strengthening existing community resources, such as traditional and religious institutions that had proven useful in other public health programs (Polio Eradication Initiative, Guinea worm elimination etc), should be strongly considered. These best practices have been well documented in Nigeria's Polio Programme and can be leveraged to improve the sensitivity of AEFI surveillance in Nigeria.

## Limitations of the study

The lack of data from respondents working at private health facilities and general hospitals is a limitation of this study. However, the contribution of private health service providers and secondary health care facilities to routine immunisation (RI) services is very limited in this study setting. Indeed, in most of the General Hospitals providing routine immunisation services in the State, the health workers are often seconded by the Local Government Health Authority (PHC level) to support RI services due to human resource deficit. Therefore, the experience is likely to be the same, and the absence of perspectives from these sources is unlikely to change the study findings. Moreover, the proportion of the General Hospitals in the State is less than 5% of the total number of RI service delivery points.

Secondly, the study is limited to one State in northwestern Nigeria. However, the State has 21 LGAs with a population of about 5 million people inhabiting the geographical space. The study could have been extended to other States, but for funding constraints and limited data collection time across a vast country like Nigeria. On the other hand, the finding of this study may largely be similar to what is obtainable in other States and regions of the country. Further, the limitation was mitigated in this study, given the interviews with or perspectives from national stakeholders in Nigeria's pharmacovigilance system.

## Conclusion

AEFI surveillance and documentation system is established but still evolving in Nigeria. However, its functionality is sub-optimal, and the capacity is yet to be fully harnessed due to some health system and socio-ecological impediments (non-health system challenges). Sadly, AEFI surveillance and documentation have not been taken as a national priority until recently, when the WHO and the NPHCDA drafted a national policy on AEFI surveillance and management in Nigeria. AEFI surveillance and documentation, in practical terms, have not been seen or considered a core component of the immunisation service delivery, especially at the operational level. However, the increasing introduction of vaccines under the EUA by WHO has further drawn attention to the relevance and dire need to strengthen AEFI surveillance in Nigeria, and other developing countries. Findings from this study provides empirical evidence and serve as an advocacy tool for vaccine pharmacovigilance strengthening in Nigeria.

Addressing the health system challenges and non-health system bottlenecks identified in this study would improve AEFI surveillance and documentation and impact the quality and effectiveness of service delivery packages at the primary health care (PHC) level. All

stakeholders must provide robust funding and logistics, and improve accountability at all levels to strengthen pharmacovigilance, restore public confidence in immunisation and promote vaccine demand.

The authors therefore recommend the following:

1. Training and supportive supervision need to be upscaled. Training health workers on AEFI and its surveillance should be extended to health training institutions in Nigeria to build their capacity and provide a baseline understanding of the AEFI- its surveillance and documentation before their professional career. This will improve their knowledge and stimulate the right attitude towards AEFI reporting and documentation at their various duty posts (health facilities or service centers). Pre-career and on-the-job-training of health workers may likely reduce the fear of reporting and mitigate the feeling of professional incompetence that deter reporting and documentation of AEFI among health workers. Further, supportive supervision during their professional life provides an on-the-job training opportunity and a means of instituting accountability for the health workers at the PHC level.

2. Prioritisation and investment in AEFI surveillance as part of the broader pharmacovigilance effort by government, pharmaceutical companies, partners, and donor agencies, especially in developing country settings, that rarely benefit from clinical trials during vaccine and drug development. The pharma should be obligated to provide special funds to support AEFI surveillance and coordination platforms (NEC and AEFI committees at State and LGA) at levels. This step should be supported by a legal framework that will facilitate funding for pharmacovigilance and clinical management of the acute presentations of AEFI and their sequelae.

3. Public Laboratory capacity at the sub-national level needs strengthening as the quality and plethora of laboratory data/investigation available to the NEC will guide their capacity to conduct a useful and competent causality assessment for the reported/investigated serious AEFI cases in a timely manner. This will facilitate prompt feedback to the health workers and the parent/givers.

4. There is a need to fast-track the consolidation and finalisation of the Med Safety App as this will improve case detection, spontaneous reporting and stimulate investigation and timely analysis and feedback.

5. Addressing socio-ecological factors, such as women education, improved access to health services, better rural infrastructure (road access and information and telecommunication services) and improve security for health workers and general population, improved flow and quality of health information through effective community linkages and engagement, would impact positively the detection, reporting, investigation, documentation of AEFI including feedback mechanism, and ultimately improve public confidence in immunisation and generate demand for vaccines

## Supporting information

**S1 Checklist. COREQ (COnsolidated criteria for REporting Qualitative research) checklist.**
(PDF)

**S1 Data. Interview transcriptions.**
(ZIP)

**S2 Data. Interview recordings—Part One.**
(ZIP)

**S3 Data. Interview recordings—Part Two.**
(ZIP)

**S4 Data. Interview recordings—Part Three.**
(ZIP)

**S5 Data. Interview recordings—Part Four.**
(ZIP)

## Acknowledgments

The authors thanked the staff of the State and National Primary Health Care Development Agencies, the National Agency for Food and Drug Administration and Control and the Development Partners who volunteered to participate in this study.

## Author Contributions

**Conceptualization:** Semeeh Akinwale Omoleke.

**Data curation:** Semeeh Akinwale Omoleke.

**Formal analysis:** Semeeh Akinwale Omoleke.

**Investigation:** Semeeh Akinwale Omoleke.

**Methodology:** Semeeh Akinwale Omoleke, Moyosola Bamidele.

**Project administration:** Laurent Cleenewerck de Kiev.

**Supervision:** Moyosola Bamidele, Laurent Cleenewerck de Kiev.

**Writing – original draft:** Semeeh Akinwale Omoleke.

**Writing – review & editing:** Semeeh Akinwale Omoleke, Moyosola Bamidele, Laurent Cleenewerck de Kiev.

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
