## [Decision Letter · Decision Letter 0]

18 Nov 2022

PGPH-D-22-01497

Type of Paper: Research Article

Full Title: Barriers to Optimal AEFI Surveillance and Documentation in Nigeria: Findings from a Qualitative Survey

Dear Dr. Omoleke 

Thank you for submitting your manuscript to PLOS Global Public Health. After careful consideration, we feel that it has merit but does not fully meet PLOS Global Public Health’s publication criteria as it currently stands. Therefore, we invite you to submit a revised version of the manuscript that addresses the points raised during the review process.

We look forward to receiving your revised manuscript.

Kind regards,

Peter Bai James, PhD

Academic Editor

Journal Requirements:

1. In the online submission form, you indicated that "The recordings and transcripts for this work are available upon request to the corresponding author.". All PLOS journals now require all data underlying the findings described in their manuscript to be freely available to other researchers, either 1. In a public repository, 2. Within the manuscript itself, or 3. Uploaded as supplementary information.

Additional Editor Comments (if provided):

1.The authors need to explain how the trustworthiness of the study was i.e., Credibility, Dependability, Confirmability and Transferability.

2. The authors should ensure that the COREQ guidelines for reporting qualitative study is adhered to. Please submit a supplementary file of COREQ guidelines, which shows that the study adhered to the guideline.

3. The authors need to state who conducted the interviews and analyzed, including his/her competency 

4. Can the authors provide basic demographics of the participants for the FGD and IDI

 5. Line 156. Please remove the z 

Reviewers' comments:

Reviewer's Responses to Questions

**Comments to the Author**

1. Does this manuscript meet PLOS Global Public Health’s publication criteria? Is the manuscript technically sound, and do the data support the conclusions? The manuscript must describe methodologically and ethically rigorous research with conclusions that are appropriately drawn based on the data presented.

Reviewer #1: Yes

Reviewer #2: Yes

2. Has the statistical analysis been performed appropriately and rigorously?

Reviewer #1: N/A

Reviewer #2: Yes

3. Have the authors made all data underlying the findings in their manuscript fully available (please refer to the Data Availability Statement at the start of the manuscript PDF file)?

Reviewer #1: No

Reviewer #2: No

4. Is the manuscript presented in an intelligible fashion and written in standard English?

Reviewer #1: Yes

Reviewer #2: Yes

5. Review Comments to the Author

Reviewer #1: This is a very interesting paper that identifies very real and practical challenges to AEFI surveillance in low resource settings.

The introduction appropriately places the problem in context.

The methods are generally sound.

The results are compelling and of interest.

The discussion places the findings in context of existing research appropriately demonstrating general consistency of the findings.

My comments:

1. The sample size is small - this would be the main limitation of the paper. The potential sample however is limited.

2. It wasnt clear how the interview guide was constructed. More information on this would be worthwhile.

3. I dont beieve any comment was made on whether saturation of themes was achieved.

4. For the results the quotes may work best in a results table.

5. For the discussion - while the results are placed in the LMIC context I dont believe WHO's extensive guidance on AEFI reporting is referenced. https://www.who.int/teams/regulation-prequalification/regulation-and-safety/pharmacovigilance/health-professionals-info/aefi

It seems like the themes identified in this and other papers referenced would be relevant to the WHO.

6, A brief mention of digital reporting of AEFI reporting was alluded to - some more description on this would be interesting.

7. It wasnt clear to me how AEFI data is used in Nigeria. Some information on data flows, and processes for handling and responding to AEFI data would be helpful. I think other sections of the discussion could be reduced to allow for this.

Reviewer #2: Abstract

- The introduction should highlight the global burden of AEFI to set the tone for the relevance of the study.

- The significance of the study is need to be well established. Indicating that the performance remains weak does not present the clear problem/ justification of the study.

- The study is a qualitative design, however, the type of qualitative design was not stated (phenomenological, ethnographic, grounded theory, historical, case study, and action research etc)

- Public health action from the study was not indicated which makes which underscores the relevance of the study

Introduction

- Line 72 - 74: the objectives are not clear. It should be measurable. The authors should conduct literature review to understand what the quantitative studies found. Since they claim no qualitative study has been conducted, this study should rather help in exploring the findings from the quantitative studies.

-Line 106: Design should reflect in abstract

- Line 123: What informed the sample size for the FGD?

- Line 127: Spell out in full for first use : NAFDAC and other abbreviations

Methods

- The methods should be aligned to the objectives of the study with sub-headings (sub-themes) for each objective. A it stands the objectives are equally not clear which makes it difficult to appreciate the linkages between the objectives and the methods.

Results

- Results

It would be good to have a table summarising the positive and negative sentiments to help the reader appreciate the findings in a snapshot.

Discussion

Sequence should be revised to follow the presentation of results to provide more clarity.

-Line 423: the word "gap" should be removed. The word inadequate is enough to indicate a negative direction of communication.

- Authors should indicate the limitations of the study

Conclusion

- Authors did not indicate the public health actions the emanated from their work which underscores the relevance of the study.

- the authors talked about addressing the challenges identified, however there are not targeted recommendations for their work and no evidence to show that their study was of any benefit to the study population

6. PLOS authors have the option to publish the peer review history of their article (what does this mean?). If published, this will include your full peer review and any attached files.

**Do you want your identity to be public for this peer review?** For information about this choice, including consent withdrawal, please see our Privacy Policy.

Reviewer #1: No

Reviewer #2: No

---

## [Decision Letter · Decision Letter 1]

7 Feb 2023

PGPH-D-22-01497R1

Type of Paper: Research Article

Full Title: Barriers to Optimal AEFI Surveillance and Documentation in Nigeria: Findings from a Qualitative Survey

Dear Dr. Omoleke,

Thank you for submitting your manuscript to PLOS Global Public Health. After careful consideration, we feel that it has merit but does not fully meet PLOS Global Public Health’s publication criteria as it currently stands. Therefore, we invite you to submit a revised version of the manuscript that addresses the points raised during the review process.

EDITOR Comments:

1. In the results sections, I suggest that you reduce some of the quotes to just one or two representative quotes.

2. Please briefly describe how trustworthiness of the findings was ensured, i.e.  how credibility, dependability, confirmability and transferability was ensured when conducting this study.  

3. Please upload a supplementary file that indicate that   you adhered to the   COREQ guidelines for qualitative research. 

Please submit your revised manuscript on or before the 20th of February 2023. If you will need more time than this to complete your revisions, please reply to this message or contact the journal office at globalpubhealth@plos.org. Please include the following items when submitting your revised manuscript:

We look forward to receiving your revised manuscript.

Kind regards,

Peter Bai James, PhD

Academic Editor

Journal Requirements:

Additional Editor Comments (if provided):

Reviewers' comments:

Reviewer's Responses to Questions

**Comments to the Author**

1. If the authors have adequately addressed your comments raised in a previous round of review and you feel that this manuscript is now acceptable for publication, you may indicate that here to bypass the “Comments to the Author” section, enter your conflict of interest statement in the “Confidential to Editor” section, and submit your "Accept" recommendation.

Reviewer #1: All comments have been addressed

Reviewer #2: All comments have been addressed

2. Does this manuscript meet PLOS Global Public Health’s publication criteria? Is the manuscript technically sound, and do the data support the conclusions? The manuscript must describe methodologically and ethically rigorous research with conclusions that are appropriately drawn based on the data presented.

Reviewer #1: Yes

Reviewer #2: Yes

3. Has the statistical analysis been performed appropriately and rigorously?

Reviewer #1: N/A

Reviewer #2: Yes

4. Have the authors made all data underlying the findings in their manuscript fully available (please refer to the Data Availability Statement at the start of the manuscript PDF file)?

Reviewer #1: Yes

Reviewer #2: No

5. Is the manuscript presented in an intelligible fashion and written in standard English?

Reviewer #1: Yes

Reviewer #2: Yes

6. Review Comments to the Author

Reviewer #1: The authors have satsifactorily addressed my comments. I would still remove at least some of the quotes from the results as some of them are also in a table. The paper does seem long and the discussion in particular. The issue of fear of reporting AEFI's seem to me the most important finding and its not addressed in the recommendations.

Reviewer #2: Data should be uploaded.

7. PLOS authors have the option to publish the peer review history of their article (what does this mean?). If published, this will include your full peer review and any attached files.

**Do you want your identity to be public for this peer review?** For information about this choice, including consent withdrawal, please see our Privacy Policy.

Reviewer #1: No

Reviewer #2: No

---

## [Editor Report · Decision Letter 2]

17 Mar 2023

Type of Paper: Research Article

Full Title: Barriers to Optimal AEFI Surveillance and Documentation in Nigeria: Findings from a Qualitative Survey

PGPH-D-22-01497R2

Dear. Omoleke

We are pleased to inform you that your manuscript 'Type of Paper: Research Article

Full Title: Barriers to Optimal AEFI Surveillance and Documentation in Nigeria: Findings from a Qualitative Survey' has been provisionally accepted for publication in PLOS Global Public Health.

Best regards,

Peter Bai James, PhD

Academic Editor